# MACS CODER: A Multi-Agent Coding Framework for Small LMs — From Fast Thinking to Deep Planning

## Abstract

Large Language Models (LLMs) have made significant strides in code generation, yet solving complex programming tasks remains a major challenge. Current state-of-the-art (SOTA) multi-agent frameworks, while powerful, often use a resource-intensive, **one-size-fits-all** strategy. We introduce **MACS-Coder** (Multi-Agent Adaptive Coding Structure), a novel dual-process framework designed for high efficiency on personal computers (macOS and Windows PCs). Inspired by human cognition, it comprises two systems: a **Fast Thinking System** for rapid, low-cost code generation, and a **Deep Planning System** for methodical, deliberative problem-solving. This dual architecture allows small models to achieve performance comparable to much larger proprietary models while consuming far less energy and producing lower $CO_2$ emissions. MACS-Coder dynamically adapts its strategy, employing its Fast Thinking System for simpler tasks and activating its Deep Planning System—composed of planning, structured templating, and fine-grained debugging agents—for complex challenges. Extensive experiments across multiple benchmarks, including the highly challenging LiveCodeBench, demonstrate that MACS-Coder achieves new SOTA pass@1 results. Using the gpt-oss-20B model, it attains accuracies of **99.4%** on HumanEval, **93.2%** on MBPP, and **83.2%** on LiveCodeBench V5, consistently outperforming prior methods such as CodeSIM and MapCoder in both accuracy and computational efficiency. When scaled to a larger open-source backbone (e.g., gpt-oss-120B), MACS-Coder achieves SOTA performance on live-coding benchmarks, surpassing earlier SOTA models. The primary contribution of our work is to bridge the performance gap between compact open-source models and elite closed-source systems: we show that an open-source gpt-oss-20B model empowered by MACS-Coder can achieve performance comparable to top-tier models such as o4-Mini (High) and Gemini 2.5 Pro. By making SOTA-level performance more accessible and resource-efficient, MACS-Coder represents a significant step toward democratizing advanced AI-assisted programming. We will open-source the framework and evaluation code to facilitate future research. Explore our code and video demo at here.

## 1 Introduction

In recent years, the rapid rise of large language models (LLMs) has driven major advances in AI-assisted programming. These models fundamentally change how developers generate, reason about, and debug code, enabling tasks such as maintaining codebases, adding new features, and even building complete applications without programming expertise, a phenomenon sometimes referred to as **"vibe coding."**

While flagship proprietary models like OpenAI's GPT-5 line (OpenAI, 2025) and Anthropic's Claude-4 (Anthropic, 2025) continue to push the state of the art in coding, a rapidly expanding ecosystem of open-source families (e.g., Meta's Llama-4 (MetaAI, 2025), Alibaba's Qwen-3 (Yang et al., 2025)) is narrowing performance gaps and broadening accessibility for researchers.

Despite these gains, two practical limitations remain. First, top-performing proprietary models typically depend on very large parameter counts and heavy compute budgets, restricting routine

access for individuals, small laboratories, and many production systems. Second, modern multi-agent and pipeline-based approaches (for instance, staged planning → coding → verification systems such as MapCoder (Islam et al., 2024a) and CodeSIM (Islam et al., 2025a) improve robustness by decomposition, but commonly adopt a **"one-size-fits-all"** strategy that invokes a complete, resource-intensive stack for every problem instance. This leads to wasted computation on simple tasks and can still be suboptimal for truly hard problems because the pipeline does not adaptively match reasoning effort or specialist models to instance difficulty.

Small models (the model parameters $\lesssim$ 20 B) remain valuable for research and personalized workflows due to their low cost and ease of fine-tuning, but often lag behind larger models in core coding and mathematical reasoning benchmarks. The central problem we address is whether it is possible to retain the reliability of cooperative multi-agent pipelines while greatly reducing average compute by tailoring the amount and kind of reasoning performed per instance.

To fill this gap, we propose **MACS-Coder**. MACS-Coder integrates three core ideas: (1) a *Fast-and-Deep Planning* dual-system architecture that dynamically selects between a low-cost fast path for easy instances and a deep-planning pipeline for harder ones; (2) *structured generation* through code templates produced by an STD-IO Tool to improve output stability and make verification easier; and (3) a *fine-grained debugging* mechanism that isolates and repairs different classes of errors precisely. The entire architecture is illustrated in Figure 1. In Figure 2, we illustrate how MACS-Coder operates during the Deep-Planning phase, highlighting its efficiency and comprehensiveness. The workflow demonstrates a reasoning process that systematically explores solutions, integrates contextual knowledge, and refines intermediate decisions.

We evaluated MACS-Coder on the challenging LiveCodeBench V5 code generation benchmark (Jain et al., 2024), with its performance on basic tasks such as HumanEval and MBPP detailed in the Appendix B.1. Our experiments primarily feature the gpt-oss-20B (Agarwal et al., 2025) and Qwen3-series (Yang et al., 2025) models, while extensive evaluations of a wider variety of open-source small language models (SLMs) can be found in Appendix B.2. The results show that MACS-Coder improves computational efficiency while matching or exceeding strong baselines such as CodeSIM and MapCoder on accuracy. In particular, our experiments demonstrate that targeted orchestration and structured guidance can substantially amplify the effective performance of midsize open models, enabling competitive results against lighter proprietary variants while dramatically reducing average compute.

## Contributions

- We analyze cost performance trade-offs across single-agent, fixed multi-agent, and adaptive orchestration strategies on a spectrum of coding tasks.

- We introduce MACS-Coder, an adaptive orchestration framework that combines instance-wise difficulty estimation, progressive delegation, and early exit decisions.

- We empirically demonstrate that MACS-Coder preserves high success rates on hard tasks while significantly reducing average computation on easy/medium tasks and that it improves the utility of small/medium open models when used as targeted specialists.

- We will release the MACS-Coder implementation and experimental configurations to support reproducibility and future research.

## 2 RELATED WORK

### 2.1 LLMs FOR CODE

Program synthesis and code generation have long been fundamental challenges in artificial intelligence. The evolution of LLMs has significantly transformed the landscape of automated code-related tasks, including code completion, code translation, code summarization, and code repair. General-purpose LLMs are pre-trained on large-scale text, code, and mathematical data to strengthen logical reasoning and code understanding.

Notable proprietary models include Anthropic's Claude series (Anthropic; 2025), OpenAI's GPT family (e.g., GPT-4 (Achiam et al., 2023) and the more recent GPT-5 line (OpenAI, 2025), including

the "o" series reasoning models (OpenAI, 2025)), and Google's Gemini series (notably Gemini 2.5 (Comanici et al., 2025)). At the same time, the open-source ecosystem has matured, producing powerful models that democratize code generation. Representative open-source examples include the Llama series and Meta's Code Llama variants (Grattafiori et al., 2024; MetaAI, 2025; Rozière et al., 2023), the Mistral family and Mistral Coder (MistralAI, 2024; Mistral AI team, 2024), DeepSeek and DeepSeek Coder (Shao et al., 2024; DeepSeek-AI et al., 2024; Guo et al., 2024), and Alibaba's Qwen series including Qwen-2.5/3 Coder variants (Yang et al., 2024; 2025; Hui et al., 2024; Qwen Team, 2025). These models have demonstrated strong capabilities on many programming problems and form the foundational core of current code-generation agents.

## 2.2 Multi-Agent Code Generation

Although LLM-based code generation techniques can produce standalone programs, their single-response mode exposes significant limitations for complex engineering-oriented development. Native LLMs cannot autonomously decompose tasks, interact with real development environments, validate generated code, or implement continuous self-correction, so they struggle with cross-file context, dynamic debugging, and iterative optimization. To address these gaps, research has shifted from prompt engineering toward multi-agent frameworks that decompose problem solving into stages handled by specialized agents.

**MapCoder** (Islam et al., 2024b) proposes a multi-agent framework that mimics the human development cycle, incorporating agents for planning, coding, and debugging. Although this approach shows progress, its workflow is fixed and lacks adaptability to problem difficulty, leading to significant resource consumption on all problems and limiting its efficiency.

**CodeSIM** (Islam et al., 2025b) introduces a novel verification method inspired by human problem-solving, using input-output simulation to validate generated plans and perform internal debugging. CodeSIM has achieved SOTA performance in accuracy, but similar to MapCoder, it employs a resource-intensive, fixed pipeline that fails to dynamically adjust its strategy based on problem complexity, leaving room for improvement in computational efficiency.

## 2.3 Code Debugging

**LDB** (LLM Debugger) (Zhong et al., 2024) is an external debugging framework that refines programs using runtime execution information. By segmenting code into basic blocks and tracking intermediate variable values, LDB enables an LLM to perform block-by-block verification against the task description, allowing it to accurately pinpoint and correct errors.

**LPW** (LLM Programming Workflow) (Lei et al., 2025) features a key innovation called "plan verification," where a natural language plan is validated against tests to generate a detailed solution with expected intermediate outputs. This allows for precise debugging by comparing a failed program's execution trace to the pre-verified outputs, creating a structured signal for refinement.

## 3 MACS-Coder

Our objective is to propose an efficient code agent through a two-stage "Fast and Deep Planning" design. The Fast-Thinking stage addresses simple problems that do not require extensive thought, while the Deep-Planning stage guides the LLM through step-by-step reasoning based on the processes of human engineers. By leveraging the concept of test-time compute, it allocates more resources to enhance the LLM's final performance. Inspired by software engineering practices, we designed a four-step LLM agent process for the Deep-Planning stage to simulate an engineer's workflow: analyzing the problem (Planning Agent), generating structured code (Coding Agent), debugging (Debugging Agent), and handling standard I/O (STD IO Tool). A detailed ablation study, presented in Appendix C, quantifies the significant impact of each of these components on both Pass@1 accuracy and token consumption. Drawing inspiration from recent works like MapCoder, CodeSIM, and LDB, we developed MACS-Coder to achieve a balance of performance and efficiency.

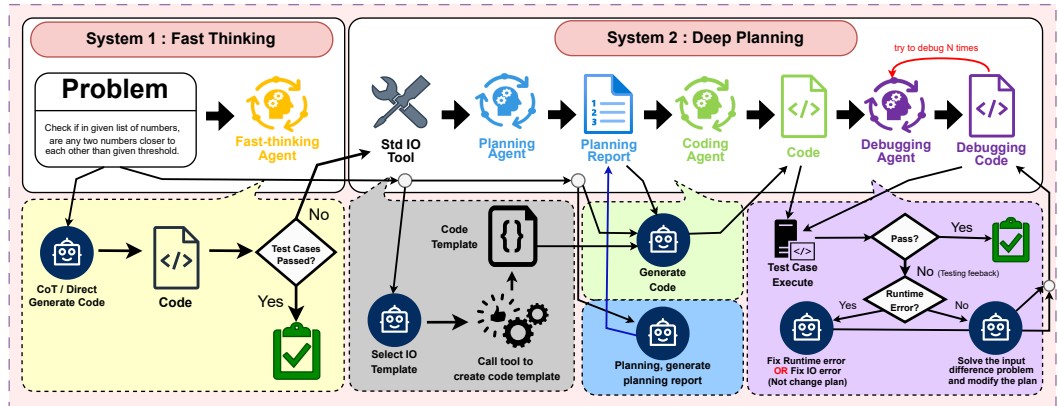

Figure 1: Overview of MACS-Coder

## 3.1 FAST THINKING SYSTEM

The initial stage of our framework is the **Fast-Thinking System**, which serves as a rapid and resource-efficient filter. This stage is designed to handle problems for which the LLMs already exhibit high proficiency. The **Fast-Thinking Agent** generates a code solution with minimal guidance and evaluates its correctness against a set of unit tests.

This process acts as a simple yet effective heuristic for difficulty assessment:

- If the generated code passes all unit tests, the solution is deemed correct and is returned immediately, minimizing computational expenditure.
- If any test fails, the problem is considered non-trivial and is escalated to the Deep-Planning System.

Our implementation adapts its prompting strategy based on the problem domain; for complex competitive programming tasks (LiveCodeBench V5), the agent attempts direct code generation. For benchmarks with clearer solution patterns (HumanEval, MBPP), a lightweight planning-style prompt is used to improve initial accuracy.

## 3.2 DEEP PLANNING SYSTEM

This stage is designed for difficult problems or those where the LLM's responses are unstable. By increasing resource consumption, it enables the LLM to think more comprehensively and perform debugging to enhance its final performance.

### 3.2.1 ENVIRONMENT PREPARATION: STD IO TOOL

The STD IO Tool is specifically designed for competitive programming problems, for which we created several input and output templates based on common formats. The LLM uses a tool-calling mechanism to select an appropriate template, ultimately producing a complete **code template**. This allows subsequent agents to modify the template directly when generating code rather than starting from scratch. Furthermore, our designed code templates improve the readability of the final code; details are provided in Appendix D.1. Note that this tool is not activated for simpler benchmarks like HumanEval and MBPP, which lack complex I/O handling. The role of the STD IO Tool within our overall workflow is illustrated in Figure 2 (highlighted as **Step 1**).

### 3.2.2 STRATEGY FORMULATION: PLANNING AGENT

In the strategy formulation stage, the Planning Agent prompts the LLM to generate a **Planning Report** from the problem description. This report outlines the core algorithm, analyzes potential edge cases, and establishes a coding plan, providing a robust foundation for the implementation phase. To select an optimal algorithm, the LLM first enumerates all viable solutions and then chooses the most

stable and straightforward strategy that prioritizes robustness and increases the likelihood of a correct solution. This stage corresponds to **Step 2** in our workflow, as depicted in Figure 2. (Full details of the instructional prompts are in Appendix D.2.)

### 3.2.3 CODE SYNTHESIS: CODING AGENT

The Coding Agent (**Step 3**, Figure 2) synthesizes the final code. It takes the problem description, the Planning Report, and the Code Template as input, generating an implementation that conforms to the template's structure. To enforce adherence to the plan, the LLM is also prompted to generate code comments corresponding to each step in the Planning Report. The resulting code is then validated against a suite of unit tests. Code that passes all tests is considered successful; otherwise, it is passed to the Debugging Agent for refinement. (The prompts for this stage are detailed in Appendix D.3 and Appendix D.4.)

### 3.2.4 AUTOMATED DEBUGGING: DEBUGGING AGENT

In the Debugging Agent stage (**Step 4**, Figure 2), the agent receives the original problem, the Planning Report, the generated code, and the execution log from unit testing as input to debug the code. The Debugging Agent consists of three modules: the STD IO Error Block, the Runtime Error Block, and the Wrong Answer Block. We first perform a string comparison on the execution log to determine if the code executed correctly. If it did, the process moves to the Output Error Block; otherwise, it is routed to either the Runtime Error Block or the STD IO Error Block. If this is the first execution failure, it enters the STD IO Error Block; subsequent failures are handled by the Runtime Error Block.

**STD IO Error Block** This block is entered on the first execution failure. Our experiments revealed that many execution failures are related to input parsing. If data is not read correctly, an error occurs. Therefore, this stage prompts the LLM to regenerate the code for reading and displaying input, while the core algorithmic code is preserved.

**Runtime Error Block** If the code fails to execute for a second time or more, this block is entered. It uses a more general prompt, asking the LLM to analyze the error itself and identify the cause of the failure to modify the code, again without altering the core problem-solving logic.

**Wrong Answer Block** If the code executes correctly but the output does not match the expected answer, this block is entered. Here, the LLM is prompted to analyze the incorrect output, simulate the failing test case, and generate an improved algorithm to modify the code.

(The specific debugging prompts for each block are presented in Appendix D.5.)

## 4 EXPERIMENTAL SETUP

### 4.1 EVALUATION BENCHMARKS AND BASELINES

We evaluate our method, MACS-Coder, on a diverse set of programming benchmarks. Our primary evaluation is on **LiveCodeBench V5** (Jain et al., 2024), a contamination-free benchmark of 880 competitive problems (279 Easy, 331 Medium, 270 Hard). For a comprehensive assessment of its performance on basic tasks, we present detailed results for **HumanEval** (Chen et al., 2021) (164 problems), **MBPP** (Austin et al., 2021) (974 problems), and their extended-test (-ET) versions (Dong et al., 2023) in Appendix B.1.

We compare MACS-Coder against several strong baselines: **Direct** generation (Chen et al., 2021), **Chain of Thought (CoT)** (Wei et al., 2022), the multi-agent **MapCoder** (Islam et al., 2024b), and a state-of-the-art debugging framework, **CodeSIM** (Islam et al., 2025b). We also include **LDB** (Zhong et al., 2024), although its use is confined to the fundamental benchmarks (HumanEval/MBPP), as its framework is incompatible with the LiveCodeBench test format.

### 4.2 IMPLEMENTATION DETAILS

Our core evaluation metric is **pass@1**. Our primary experiments utilize three models: **Qwen3 (8B, 14B)** and **gpt-oss (20B)**. To validate generalization, we present further experiments on a broader

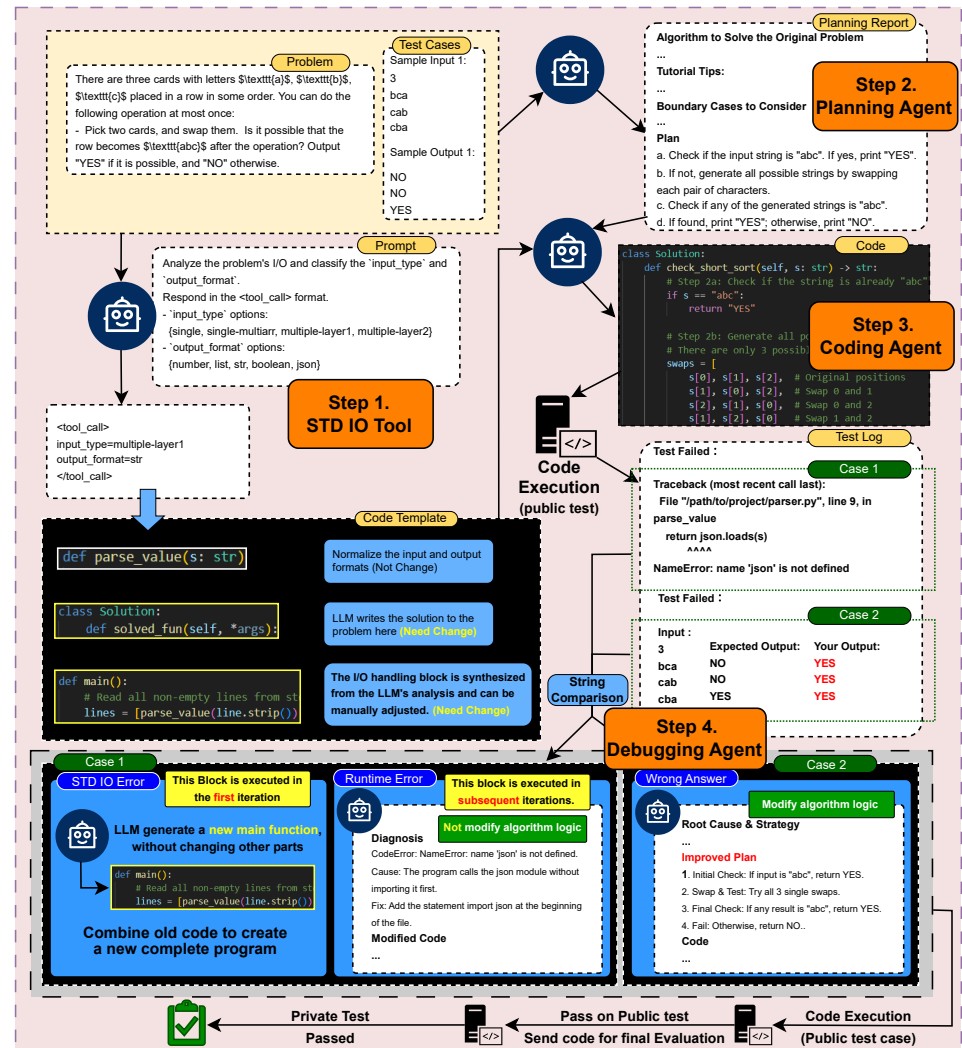

Figure 2: This figure illustrates the MACS-Coder pipeline in the Deep-Planning stage, featuring structured generation for stable, verifiable outputs and fine-grained debugging for precise error isolation, jointly improving efficiency and robustness in code generation.

range of models, including **Llama3.1-8B**, **Gemma2-9B**, **Mistral-8B**, and **Qwen2.5-Coder-7B**, in Appendix B.2. We contextualize our results against top proprietary models from the official **LiveCodeBench Leaderboard**.

Our MACS-Coder framework is governed by two key hyperparameters that control its nested-loop structure. These are $p$, the maximum number of outer *planning cycles*, and $d$, the maximum number of inner *debug attempts* per planning cycle. Key values for these hyperparameters and model-specific configurations used to suppress unwanted internal reasoning are detailed in Table 1.

Table 1: Hyperparameter and model-specific configurations.

| Model Category | Parameters $(p, d)$ | Configuration Detail |
|---|---|---|
| Primary Models | $p = 5, d = 5$ | Qwen3: "Non-Thinking Mode" + specific prompt (Ma et al., 2025). gpt-oss: "Reasoning:Low" prompt (OpenAI, 2025). |
| Generalization Models | $p = 1, d = 5$ | N/A |

**Contest-Level Problems (LiveCodeBench V5)**

| LLM | Approach | Easy | | | Medium | | | Hard | | | Total |
|---|---|---|---|---|---|---|---|---|---|---|---|
| | | ACC (%) | Time (s) | Tokens | ACC (%) | Time (s) | Tokens | ACC (%) | Time (s) | Tokens | ACC (%) |
| gpt-oss 20B | Direct | 87.1 | 3.66 | 883 | 71.3 | 13.24 | 3,173 | 42.2 | 27.58 | 6,580 | 67.3 |
| | CoT | 87.8 | 3.07 | 721 | 68.3 | 10.48 | 2,465 | 38.9 | 23.59 | 5,459 | 65.4 |
| | MapCoder | 95.7 | 19.86 | 4,762 | 80.7 | 40.09 | 9,505 | 55.9 | 150.42 | 35,148 | 77.8 |
| | CodeSIM | 97.1 | 13.82 | 3,279 | 81.9 | 40.55 | 9,683 | 61.5 | 176.13 | 41,380 | 80.4 |
| | **MACS-Coder** | **97.5** | **5.06** | **1,204** | **87.9** | **23.83** | **5,690** | **63.0** | **143.56** | **33,968** | **83.2** |
| Qwen3-14B | Direct | 67.4 | 28.38 | 1,799 | 38.4 | 90.69 | 5,660 | 25.2 | 134.05 | 8313 | 43.5 |
| | CoT | 58.1 | 30.64 | 1,911 | 36.9 | 97.24 | 6,001 | 23.0 | 148.54 | 9,063 | 39.3 |
| | MapCoder | 87.8 | 82.82 | 5,144 | 65.0 | **260.51** | **15,831** | 32.6 | **426.70** | **25,833** | 62.2 |
| | CodeSIM | 91.8 | 142.54 | 8,622 | 61.6 | 467.18 | 28,107 | 28.5 | 726.95 | 44,037 | 61.0 |
| | **MACS-Coder** | **95.3** | **64.69** | **3,971** | **72.8** | 308.64 | 18,586 | **35.9** | 768.47 | 46,220 | **68.6** |
| Qwen3-8B | Direct | 61.3 | 32.46 | 2,072 | 39.0 | 51.57 | 3,231 | 20.7 | 60.68 | 3,748 | 40.4 |
| | CoT | 47.7 | 4.30 | 259 | 13.0 | 16.52 | 1,001 | 4.8 | 35.72 | 2,145 | 21.4 |
| | MapCoder | 72.0 | **39.03** | **2,475** | 31.1 | **82.07** | **5,183** | 10.7 | **134.31** | **8,356** | 52.0 |
| | CodeSIM | 87.1 | 89.34 | 5,688 | 53.2 | 262.34 | 16,592 | 22.2 | 678.91 | 41,591 | 54.4 |
| | **MACS-Coder** | **94.6** | 49.71 | 3,130 | **66.5** | 150.07 | 9,331 | **26.7** | 319.72 | 19,716 | **63.1** |

Table 2: Performance comparison of various methods on the LiveCodeBench V5 benchmark, utilizing state-of-the-art (SOTA) open-source SLMs including gpt-oss-20B, Qwen3-14B, and Qwen3-8B. We evaluate each approach based on three key metrics: ACC (Pass@1 Accuracy), Time (average seconds per problem), and Tokens (average generated response tokens per problem). Our proposed method, MACS-Coder, consistently achieves the highest accuracy across all tested models.

## 5 RESULTS

In Table 2, we evaluate the model's performance on complex, competitive-level programming tasks. MACS-Coder demonstrates a significant superiority over all baseline methods in solving these complex problems. With the most powerful model, gpt-oss-20B, MACS-Coder achieves a total accuracy of 83.2%, setting a new state-of-the-art (SOTA) record and clearly surpassing the next-best methods, CodeSIM (80.4%) and MapCoder (77.8%).

Beyond top-tier accuracy, another core advantage of MACS-Coder lies in its exceptional computational efficiency. The data shows that MACS-Coder achieves higher accuracy while consuming far fewer computational resources (time and tokens) than its competitors. For instance, on the gpt-oss-20B model, MACS-Coder is not only the most accurate but also saves nearly 18% in token consumption on "Hard" level problems compared to CodeSIM (33,968 vs. 41,380). This efficiency advantage is due to our "fast-thinking" system, which quickly handles simple problems, strategically reserving computational resources for challenges that genuinely require deep thought.

The effectiveness of MACS-Coder remains consistent across models of different scales. Particularly noteworthy is its profound empowering effect on smaller models. The Qwen3-8B model equipped with MACS-Coder reaches a total accuracy of 63.1%, not only far exceeding other methods at the same scale but also outperforming the larger Qwen3-14B model when using MapCoder (62.2%). These results strongly indicate that MACS-Coder's "Fast and Deep Planning" dual-system framework marks a significant advancement in handling the complexity of competitive-level problems.

**Comparison with State-of-the-Art Models** To further validate the absolute competitiveness of the MACS-Coder framework, we compared it with the current strongest open-source and proprietary models on the LiveCodeBench benchmark. The data for this comparison is sourced from the official LiveCodeBench leaderboard (Jain et al.), covering contest problems from **May 1, 2023, to February 1, 2025**. As shown in Table 3, the results indicate that MACS-Coder can elevate the capabilities of open-source small models to a level competitive with top-tier proprietary models.

The most striking result is that the gpt-oss-20B model equipped with MACS-Coder achieves a Pass@1 score of 83.2%. This performance not only surpasses powerful proprietary models like Grok-3-Mini (81.4%) and o3-Mini (80.5%) but is also within a close margin of Gemini-2.5 Pro (84.7%). This demonstrates that MACS-Coder is an extremely powerful performance enhancer, enabling a 20B-level open-source model to compete with the industry's top models.

Furthermore, the value of MACS-Coder lies in its ability to enable smaller models to achieve SOTA-level performance, making top-tier capabilities more accessible and deployable. The Qwen3 14B with MACS-Coder (68.6%) easily surpasses the un-enhanced gpt-oss-20b (67.3%) and is on par with o1-Mini (68.4%). Even the smaller Qwen3 8B, empowered by MACS-Coder (63.1%), outperforms a range of powerful models like DeepSeek V3 (56.3%) and Claude-3.5-Sonnet (51.5%). These results strongly demonstrate that the MACS-Coder framework provides an efficient pathway for the open-source community to tackle complex programming challenges that were previously only manageable by top-tier proprietary models, using relatively smaller models.

Table 3: Effectiveness of **MACS-Coder** on Live-CodeBench V5 from **May 1, 2023, to February 1, 2025**. Models enhanced with our method show substantial Pass@1 score improvements. The best result in each column is in **bold** and the second best is underlined.

| LLM | LiveCodeBench V5 | | | |
| --- | --- | --- | --- | --- |
| | Easy | Medium | Hard | Pass@1 |
| **gpt-oss-120b (MACS-Coder)** | 97.9 | 90.6 | **70.7** | **86.8** |
| o4-Mini (High) | 98.2 | 89.7 | 67.4 | 85.6 |
| Gemini-2.5 Pro | 98.2 | **91.8** | 61.9 | 84.7 |
| **gpt-oss-20b (MACS-Coder)** | 97.5 | 87.9 | 63.0 | 83.2 |
| Grok-3-Mini (High) | 98.6 | 88.8 | 54.4 | 81.4 |
| gpt-oss-120b | 93.2 | 85.5 | 62.2 | 80.8 |
| o3-Mini (High) | **98.9** | 88.5 | 51.5 | 80.5 |
| o1 (Med) | **98.9** | 84.6 | 49.3 | 78.3 |
| DeepSeek-R1-Preview | **98.9** | 84.8 | 47.7 | 77.9 |
| **Qwen3 14B (MACS-Coder)** | 95.3 | 72.8 | 35.9 | 68.6 |
| O1-Mini | 95.0 | 73.4 | 34.8 | 68.4 |
| gpt-oss-20b | 87.1 | 71.3 | 42.2 | 67.3 |
| **Qwen3 8B (MACS-Coder)** | 94.6 | 66.5 | 26.7 | 63.1 |
| DeepSeek V3 | 90.8 | 59.4 | 17.0 | 56.3 |
| o1-Preview | 94.3 | 56.8 | 14.1 | 55.6 |
| Claude-3.5-Sonnet | 92.9 | 46.3 | 14.9 | 51.5 |
| Qwen3 14B | 67.4 | 38.4 | 25.2 | 43.5 |
| GPT-4o | 87.4 | 36.8 | 6.1 | 43.4 |
| Gemini-Pro-1.5 | 87.2 | 31.1 | 8.9 | 42.1 |
| Qwen3 8B | 61.3 | 39.0 | 20.7 | 40.4 |

## 6 ANALYSES

### 6.1 PERFORMANCE-COST ANALYSIS

To visually demonstrate the efficiency advantage of MACS-Coder over the current SOTA method, CodeSIM, we plotted the relationship between accuracy and token consumption (as shown in Figure 3). The graph clearly indicates that MACS-Coder achieves both higher accuracy and lower token consumption across all benchmarks and models, establishing its significant superiority in performance efficiency.

The arrows in the figure consistently point from CodeSIM to MACS-Coder, always moving towards the top-left representing higher accuracy (y-axis) and lower token consumption (x-axis). This improvement is particularly pronounced in complex tasks. For example, on the LiveCodeBench benchmark, MACS-Coder not only increases the accuracy of Qwen3 8B by 8.7% but also saves over 10,000 tokens. Even with a more powerful model like gpt-oss 20B, MACS-Coder still improves accuracy by 2.8% while saving over 4,400 tokens. This comprehensive performance advantage validates the design of our "Fast and Deep Planning" framework. It intelligently allocates computational resources, avoiding unnecessary deep thought on simple problems to concentrate resources on tackling difficult ones. This makes MACS-Coder not only a more accurate solution but also a more economical and efficient framework for practical applications.

### 6.2 ANALYSIS OF PLANNING ITERATIONS

To analyze the relationship between planning-stage investment and final performance, we evaluated the Pass@1 accuracy of MACS-Coder against CodeSIM over varying planning iterations (Figure 4). The results reveal a significant efficiency advantage for MACS-Coder, which consistently outperforms CodeSIM across all tested models and iteration counts. This efficiency gain is so substantial that MACS-Coder with the Qwen3 8B model at a single iteration achieves 56.8% Pass@1, surpassing the larger Qwen3 14B model using CodeSIM (49.4%). This finding demonstrates that our framework offers significant "capability compression," enabling smaller models to achieve results previously

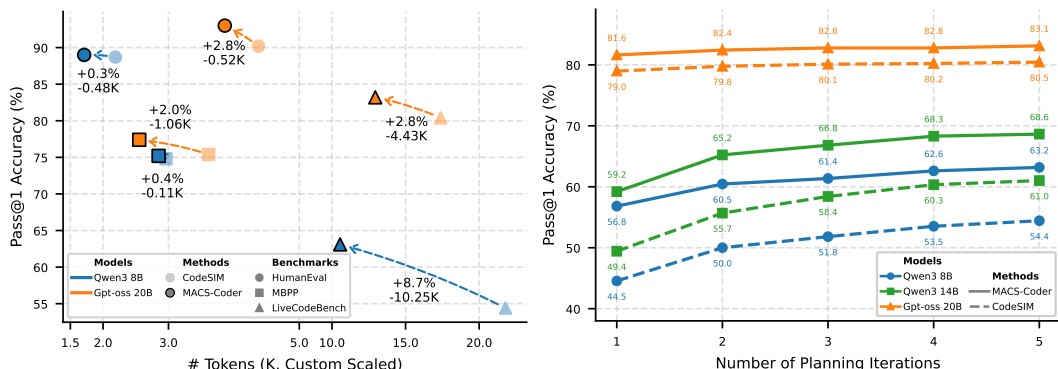

Figure 3: Accuracy-efficiency improvement achieved by our proposed methods. The plot shows that our approaches consistently push the models towards the ideal top-left corner (higher accuracy, fewer tokens).

Figure 4: Performance comparison between **MACS-Coder** (solid lines) and CodeSIM (dashed lines) across multiple planning iterations. While both methods benefit from more iterations, MACS-Coder consistently operates at a higher performance level.

requiring larger ones. This holds substantial implications for reducing computational costs while achieving state-of-the-art performance in practical applications.

## 7 CONCLUSION AND FUTURE WORK

In this paper, we introduced MACS-Coder, a novel framework designed to solve complex programming tasks. At its core, MACS-Coder features an innovative "Fast and Deep Planning" dual-system architecture that adaptively adjusts its strategy based on problem difficulty and integrates structured code template generation with a fine-grained debugging agent. Evaluation results across multiple mainstream code generation benchmarks show that MACS-Coder significantly surpasses existing SOTA methods in both accuracy and computational efficiency. Our research demonstrates that by empowering small open-source models, MACS-Coder successfully elevates their performance to a level competitive with top-tier proprietary models.

Future work will focus on extending this framework to other domains requiring complex reasoning, such as mathematical problem-solving and general question-answering, to broaden its scope and impact.

## 8 LIMITATIONS

Despite the strong performance and efficiency of MACS-Coder, some limitations exist. First, while our framework demonstrates strong generalization across multiple open-source SLMs, the magnitude of performance improvement is not entirely consistent. This variability is partly tied to the intrinsic capabilities of the base models themselves; an agentic framework can structure and guide an SLM's reasoning, but its effectiveness is ultimately constrained by the model's fundamental coding ability. If a model's baseline performance is below a certain threshold, the framework's capacity to elicit further gains is limited. Additionally, adapting our structured prompts for specific model architectures remains an area for optimization. Second, while MACS-Coder is more token-efficient than prior SOTA methods, its computational cost remains higher than Direct Prompting, marking an avenue for future work. Finally, the efficacy of our debugging agent is highly dependent on the quality and coverage of the provided test cases.

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

## A  PSEUDOCODE OF THE MACS-CODER FRAMEWORK

---

**Algorithm 1** MACS-Coder Framework

---

1: **Input:** problem $P$, unit tests $U_{tests}$
2: **Output:** solution code $C_{sol}$
3: $p \leftarrow$ max number of planning cycles
4: $d \leftarrow$ max number of debugging steps per cycle
                                                                ▷ **Stage 1: Fast Thinking System**
5: $C_{fast} \leftarrow$ FastThinkGenerate($P$)
6: $passed, log \leftarrow$ Test($C_{fast}, U_{tests}$)
7: **if** $passed$ **then**
8:     **return** $C_{fast}$
9: **end if**
                                                             ▷ **Stage 2: Deep Planning System**
10: **for** $i \leftarrow 1$ to $p$ **do**                                       ▷ Start of Planning Cycle
11:     $T_{code} \leftarrow$ GenerateCodeTemplate($P$)                      ▷ STD IO Tool
12:     $R_{plan} \leftarrow$ GeneratePlan($P$)                         ▷ Planning Agent
13:     $C_{current} \leftarrow$ GenerateCode($P, R_{plan}, T_{code}$)           ▷ Coding Agent
14:     first_failure_in_cycle $\leftarrow$ true
15:     **for** $j \leftarrow 1$ to $d$ **do**                  ▷ Start of Debugging Sub-Cycle
16:         $passed, log \leftarrow$ Test($C_{current}, U_{tests}$)
17:         **if** $passed$ **then**
18:             **return** $C_{current}$
19:         **end if**
20:         $error\_type \leftarrow$ AnalyzeLog($log$)          ▷ Determine error type
21:         **if** $error\_type =$ RUNTIME_ERROR **then**
22:             **if** first_failure_in_cycle **then**
23:                 $C_{current} \leftarrow$ FixSTDIOError($P, C_{current}$)
24:                 first_failure_in_cycle $\leftarrow$ false
25:             **else**
26:                 $C_{current} \leftarrow$ FixRuntimeError($P, C_{current}, log$)
27:             **end if**
28:         **else if** $error\_type =$ WRONG_ANSWER **then**
29:             $C_{current} \leftarrow$ FixLogicalError($P, R_{plan}, C_{current}, log$)
30:         **end if**
31:     **end for**                                ▷ End of Debugging Sub-Cycle
32: **end for**                                      ▷ End of Planning Cycle
33: **return** $C_{current}$                         ▷ Return the last generated code

---

## B  DETAILED PERFORMANCE ON FOUNDATIONAL BENCHMARKS AND GENERALIZATION MODELS

### B.1  BASIC CODE GENERATION

In the basic code generation tasks shown in Table 4, the core advantage of MACS-Coder is its outstanding overall performance efficiency, achieving an optimal balance between top-tier accuracy and computational cost.

The efficient design of MACS-Coder allows it to reach SOTA-level accuracy with fewer resources. For example, on HumanEval, the Qwen3-8B model with MACS-Coder not only achieves a top average accuracy of 89.0% but also has significantly lower time (26.80s) and token (1,711) consumption compared to the similarly performing CodeSIM.

More importantly, this high efficiency enables smaller models to achieve performance beyond their scale. The Qwen3-8B with MACS-Coder on HumanEval (89.0%) performs on par with the larger Qwen3-14B using the CoT method (89.0%). This proves that our framework can achieve results with

**Basic Programming Problems**

| LLM | Approach | HumanEval | | | | | MBPP | | | | |
|-----|----------|-----------|---|---|---|---|------|---|---|---|---|
| | | ACC (%) | Time (s) | Tokens | ET ACC (%) | Avg ACC (%) | ACC (%) | Time (s) | Tokens | ET ACC (%) | Avg ACC (%) |
| gpt-oss 20B | Direct | 62.2 | 4.07 | 981 | 55.5 | 58.8 | 47.6 | 3.51 | 848 | 31.2 | 39.4 |
| | CoT | 98.2 | 3.49 | 829 | 85.4 | 91.8 | 77.6 | 2.15 | 522 | 52.6 | 65.1 |
| | LDB | 98.8 | **3.91** | **937** | 86.0 | 92.4 | 90.7 | **4.53** | **1096** | 58.7 | 74.7 |
| | MapCoder | 97.0 | 22.31 | 5314 | 84.8 | 90.9 | 92.9 | 21.04 | 5047 | 61.2 | 77.0 |
| | CodeSIM | 97.0 | 18.43 | 4377 | 83.5 | 90.2 | 91.4 | 15.03 | 3616 | 59.4 | 75.4 |
| | MACS-Coder | **99.4** | 16.23 | 3861 | **86.6** | **93.0** | **93.2** | 10.69 | 2553 | **61.7** | **77.4** |
| Qwen3-14B | Direct | 48.2 | 14.16 | 910 | 43.3 | 45.7 | 40.1 | 14.16 | 911 | 30.0 | 35.0 |
| | CoT | 93.9 | 15.86 | 1000 | **84.1** | 89.0 | 76.6 | 14.25 | 912 | 51.6 | 64.1 |
| | LDB | 95.1 | **23.21** | **1465** | **84.1** | 89.6 | 90.9 | **36.09** | **2263** | **62.5** | **76.7** |
| | MapCoder | 94.5 | 46.16 | 2896 | 81.7 | 88.1 | 88.9 | 79.38 | 4851 | 59.2 | 74.0 |
| | CodeSIM | 95.1 | 89.62 | 5460 | **84.1** | 89.6 | **92.4** | 88.72 | 5413 | 59.7 | 76.0 |
| | MACS-Coder | **95.7** | 44.87 | 2812 | **84.1** | **89.9** | 90.7 | 67.33 | 4181 | 59.9 | 75.3 |
| Qwen3-8B | Direct | 48.8 | 25.23 | 1621 | 41.5 | 45.1 | 20.4 | 23.21 | 1500 | 14.6 | 17.5 |
| | CoT | 79.3 | 1.54 | 96 | 70.1 | 74.7 | 74.1 | 1.42 | 92 | 51.9 | 63.0 |
| | LDB | 88.4 | **11.80** | **758** | 78.0 | 83.2 | 84.1 | **12.07** | **775** | 55.2 | 69.6 |
| | MapCoder | 91.5 | 26.29 | 1685 | 78.0 | 84.7 | 86.1 | 29.57 | 1746 | 56.2 | 71.1 |
| | CodeSIM | **94.5** | 35.09 | 2188 | 82.9 | 88.7 | 89.2 | 47.68 | 2962 | **60.5** | 74.8 |
| | MACS-Coder | 93.9 | 26.80 | 1711 | **84.1** | **89.0** | **89.9** | 48.24 | 2849 | **60.5** | **75.2** |

Table 4: Performance comparison on the HumanEval and MBPP benchmarks using open-source models (gpt-oss-20B, Qwen3-14B, and Qwen3-8B). The evaluation metrics include: ACC (Pass@1 Accuracy), Time (average seconds per problem), Tokens (average generated response tokens per problem), and ET ACC (a more stringent evaluation with additional private test cases). The results show that MACS-Coder not only achieves the highest accuracy but also demonstrates superior efficiency by consuming fewer Tokens than other high-performing methods like CodeSIM and MapCoder.

more economical smaller models that previously required more expensive larger models, offering a solution for the code generation field that combines top-tier capability with practical utility.

## B.2 PERFORMANCE ACROSS OPEN-SOURCE SLMS

To further demonstrate the generalization capability of the MACS-Coder framework, we evaluated its performance across several mainstream open-source SLMs, including Llama3.1-8B, Gemma2-9B, Ministral-8B, and Qwen2.5-Coder-7B. As shown in Table 5, the results highlight MACS-Coder's exceptional adaptability and its superior balance between accuracy and efficiency.

On the more complex **LiveCodeBench V5** dataset, MACS-Coder consistently and significantly outperforms CodeSIM in accuracy across all tested models. For instance, with **Ministral-8B**, MACS-Coder achieves an accuracy of 20.0%, a substantial improvement over CodeSIM's 14.0%. Similar advantages are observed on **Llama3.1-8B** (15.1% vs. 13.9%) and **Gemma2-9B** (12.5% vs. 10.3%), proving its robust problem-solving capabilities.

On the **HumanEval** dataset, MACS-Coder demonstrates a more nuanced but equally compelling advantage. While its accuracy is highly competitive—and even superior on models like **Ministral-8B** (85.4% vs. 79.3%)—its primary strength lies in its remarkable efficiency. For example, on **Qwen2.5-Coder-7B**, although CodeSIM's accuracy is slightly higher (87.8% vs. 86.6%), MACS-Coder completes the task **29% faster** while consuming **25% fewer tokens**. This trend of achieving comparable or higher accuracy with significantly less computational cost is a consistent theme across the models.

These results strongly indicate that MACS-Coder's dual-system framework is not just optimized for a specific model but can be widely adapted to enhance different language model architectures.

Table 5: Performance comparison on HumanEval and LiveCodeBench V5 across different open-source LLMs.

| LLM | Approach | HumanEval | | | | LiveCodeBench V5 | | |
|---|---|---|---|---|---|---|---|---|
| | | ACC (%) | Time (s) | Tokens | ET ACC (%) | ACC (%) | Time (s) | Tokens |
| Llama3.1 8B Instruct | Direct | 50.0 | 5.58 | 395 | 43.3 | 10.7 | 6.51 | 451 |
| | CodeSIM | **82.3** | 69.12 | 4650 | **71.3** | 13.9 | 94.94 | 6304 |
| | MACS-Coder | 80.5 | **56.92** | **3881** | 67.7 | **15.1** | **63.69** | **4271** |
| Gemma2 9B Instruct | Direct | 66.5 | 8.98 | 477 | 58.5 | 12.0 | 10.54 | 537 |
| | CodeSIM | 80.5 | 87.35 | **3981** | 67.1 | 10.3 | **84.09** | **3870** |
| | MACS-Coder | **81.1** | **83.00** | 4029 | **67.7** | **12.5** | 110.01 | 5165 |
| Ministral 8B Instruct | Direct | 74.4 | 6.41 | 447 | 64.6 | 13.9 | 8.64 | 579 |
| | CodeSIM | 79.3 | 33.29 | 2161 | 67.7 | 14.0 | 85.40 | 5214 |
| | MACS-Coder | **85.4** | **27.00** | **1760** | **71.3** | **20.0** | **77.78** | **4803** |
| Qwen2.5-Coder 7B Instruct | Direct | 70.1 | 3.31 | 233 | 62.8 | 18.5 | 6.08 | 426 |
| | CodeSIM | **87.8** | 25.07 | 1661 | **76.8** | 23.0 | 74.85 | 5207 |
| | MACS-Coder | 86.6 | **17.70** | **1246** | 75.0 | **23.3** | **64.09** | **4427** |

Its ability to deliver high accuracy while drastically improving computational efficiency validates MACS-Coder as a universal and highly practical enhancement framework for code generation.

## C ABLATION STUDY OF MACS-CODER COMPONENTS

### C.1 EFFECTIVENESS OF THE DUAL-SYSTEM FRAMEWORK

To validate the value of our "Fast and Deep Planning" dual-system architecture, we conducted a critical ablation study with results shown in Table 6. The findings demonstrate a decisive victory for the complete framework: enabling the fast-thinking system not only saves **16.5%** in token consumption but, critically, boosts Pass@1 accuracy from 52.4% to 63.1%. This seemingly counter-intuitive accuracy gain powerfully validates our dual-system philosophy.

We attribute this phenomenon to the avoidance of "overthinking," where the sophisticated "Deep-Planning" agentic workflow, designed for complex tasks, can introduce unnecessary failure points or negative guidance when applied to simpler problems. The "fast-thinking" system acts as an intelligent triage mechanism that circumvents this risk by leveraging the model's raw capabilities on straightforward tasks. This result strongly proves that our dual-system design is not merely a layering of processes but an intelligent resource allocation strategy that raises the problem-solving ceiling by applying the right cognitive tool to the right problem, optimizing for both accuracy and computational cost.

Table 6: Token consumption analysis with and without Fast Thinking. (on LiveCodeBench V5)

| Fast Thinking | Pass@1 | Tokens | Tokens Saved (%) |
|---|---|---|---|
| ✗ | 52.4% | 12,650 | - |
| ✓ | 63.1% | 10,551 | 16.5% |

### C.2 CONTRIBUTION OF EACH AGENT IN DEEP-PLANNING SYSTEM

To quantify the contribution of the core agents in the "Deep-Planning" system, we conducted further ablation studies, with results shown in Table 7. The data clearly reveals the critical impact of the STD IO Tool and the Debugging Agent on final performance. When we remove the Debugging Agent, the Pass@1 accuracy drops from 63.1% to 55.3%, a performance loss of 7.8%. Removing the STD

IO Tool leads to an even greater performance decline, with accuracy falling to 53.6%, a drop of 9.5%. If both are disabled, the accuracy plummets to 47.8%. This means that each agent contributes approximately 8-9% to the accuracy, which, in a large test set, is equivalent to successfully solving 70 to 80 additional problems. This result strongly demonstrates that our structured template generation and fine-grained debugging processes, designed to mimic human engineers, are the two pillars that enable MACS-Coder to stably solve complex problems.

Table 7: Ablation study on STD IO Tool and Debugging Agent. (on LiveCodeBench V5)

| STD IO Tool | Debugging Agent | Pass@1 | Performance Drop |
|:---:|:---:|:---:|:---:|
| ✗ | ✗ | 47.8% | 15.3% |
| ✓ | ✗ | 55.3% | 7.8% |
| ✗ | ✓ | 53.6% | 9.5% |
| ✓ | ✓ | 63.1% | - |

## C.3    ANALYSIS OF FINE-GRAINED DEBUGGING MODULES

To validate the value of each specialized module within our fine-grained Debugging Agent, we conducted an ablation study, with results shown in Table 8. The data indicates that every error-handling module makes an indispensable contribution to the final performance. The module for handling **Runtime Errors** has the most significant impact; its removal leads to a 5.0% performance drop, highlighting the importance of managing runtime exceptions in complex programming. In comparison, the modules for handling **STD IO Errors** and **Wrong Answer** have a smaller but still crucial impact (0.5% and 0.9%, respectively), and their presence collectively ensures the integrity and robustness of the debugging process. This result proves that our fine-grained design, which allows the agent to apply the most appropriate correction strategy for different failure reasons, is key to its efficient debugging capabilities.

Table 8: Ablation study of Debugging Agent components. (on LiveCodeBench V5)

| Runtime Error | STD IO Error | Wrong Answer | Pass@1 | Performance Drop (%) |
|:---:|:---:|:---:|:---:|:---:|
| ✓ | ✓ | ✓ | 63.1% | - |
| ✗ | ✓ | ✓ | 58.1% | 5.0% |
| ✓ | ✗ | ✓ | 62.6% | 0.5% |
| ✓ | ✓ | ✗ | 62.2% | 0.9% |

## D    THE MULTI-STAGE CHAT-TEMPLATE DESIGN OF MACS-CODER

### D.1    THE PROMPT OF THE STD IO TOOL IN THE DEEP PLANNING SYSTEM

---

**The prompt of the STD IO Tool in the Deep Planning System (In Step 1)**

**System Prompt:** You are an automated code analysis tool. Based on the following problem's input and output examples, determine which "input type" and "output format" should be used. Use the `<tool_call>` tag and output **only** this:

```
    <tool_call>
  input_type = {single | single-multiarr | multiple-layer1 | multiple-layer2}
    output_format = {number | list | str | boolean | json}
    </tool_call>
```

---

**Classification rules:**

1. **single** – Only one test case provided per execution (with variants):
   - *single-simple*: a single plain input (no extra parameters)
   - *single-multiarr*: matrix or multiple-row array, for example:

     ```
     4 2
     1 2 3 4
     4 3 1 2
     ```

2. **multiple** – Multiple test cases in one run (with T first, followed by per-case data):
   - *multiple-layer1*: T lines, each a single-line test case, e.g.:

     ```
     6
     abc
     acb
     ...
     ```

   - *multiple-layer2*: multiple test cases, each spanning multiple lines, e.g.:

     ```
     4
     4
     2 2 1 2
     ...
     ```

**Output format** must be one of:

- *number*: an integer or floating-point number
- *list*: space-separated items
- *str*: a single string
- *boolean*: true or false
- *json*: valid JSON structure (e.g. {"a":1,"b":2})

---

**Example 1**

Input:
{"input":"4 2\n1 2 3 4\n4 3 1 2\n", "output":"1 4\n"}
Expected response:

```
<tool_call>
input_type = single-multiarr
output_format = list
</tool_call>
```

---

**Example 2**

Input:
{"input":"4\n4\n2 2 1 2\n3\n0 1 2\n5\n4 3 2 3 4\n9\n9 9 9 9 9 9 9 9 9\n", "output":"16\n2\n432\n430467210\n"}
Expected response:

```
<tool_call>
input_type = multiple-layer2
output_format = number
</tool_call>
```

**Example 3**
Input:
{"input": "[1, 4, 3, 8, 5]", "output": "[1, 3]" }
Expected response:

```
<tool_call>
input_type = single
output_format = list
</tool_call>
```

**Example 4**
Input:
{"input": "\"51230100\"", "output": "\"512301\""}
Expected response:

```
<tool_call>
input_type = single
output_format = str
</tool_call>
```

**Important:** Analyze the problem's intended function signature, not just the test cases. Determine the correct input_type and output_format for the function you'll implement. Below is the problem description:
{Problem}

## D.2 THE PROMPT OF THE PLANNING AGENT IN THE DEEP PLANNING SYSTEM

The prompt of the Planning Agent in the Deep Planning System. (In Step 2)

You are a programmer tasked with generating an appropriate plan to solve a given problem using the **{language}** programming language.

PROBLEM

{problem}
**Expected Output:**
Your response must be structured as follows:

PROBLEM UNDERSTANDING

- Think about the original problem. Develop an initial understanding of the problem.

PROBLEM SETTER PERSPECTIVE

Analyze the problem as if you are the problem setter:

- What core concept or algorithm is being tested?
- What kind of solution do you expect the solver to come up with?
- Why might this problem have been designed in this particular way?

POSSIBLE SOLUTION METHODS

- List **all possible algorithms or techniques** that could be applied to solve the problem.
- For each method, briefly describe:
  - Its general idea
  - Its complexity
  - Its pros and cons in the context of this problem

CHOSEN METHOD: SIMPLICITY AND RELIABILITY

- Based on the methods listed above, choose the one that is simplest, most robust, and most likely to succeed given the problem constraints.
- Justify why this method is chosen over the others.

RECALL EXAMPLE PROBLEM

Recall a relevant and distinct problem (different from the one mentioned above):

- Describe it
- Write the {language} code step-by-step to solve that problem
- Discuss the algorithm used in that example
- Generate a plan to solve that example problem

ALGORITHM TO SOLVE THE ORIGINAL PROBLEM

- Describe the chosen algorithm again, but now in the specific context of the original problem.
- Include tutorial-style tips:
  - How to approach this type of algorithm
  - Important pitfalls to avoid
  - Any tricks to make implementation easier

BOUNDARY CASES TO CONSIDER

- Based on the problem description and constraints, list all boundary cases that need special handling.
- Use clear natural language to describe these boundary cases.

PLAN

- Write down a detailed, step-by-step plan to solve the **original problem**.

- - - - - - - - - - - - - - - - - - - - - - - - - - - - - - - - - - - - - - - - - - -

**Important Instruction:**

- Strictly follow the instructions.
- Do not output any code.
- The plan must be written in natural language, not code.

## D.3 THE PROMPT OF THE CODE TEMPLATE IN THE DEEP PLANNING SYSTEM

The prompt of the Code Template in the Deep Planning System. (In Step 3)

CODE FORMAT TEMPLATE

```python
import sys, json
from typing import List, Union

# Universal input parser: automatically parse JSON, int, float, space-separated
    list, etc.
def parse_value(s: str) -> Union[int, float, str, list, dict, bool]:
    s = s.strip()
    # 1. Try JSON parsing (handles list, dict, quoted string, boolean, number)
    try:
        return json.loads(s)
    except json.JSONDecodeError:
        pass
    # 2. Integer detection
    if s.isdigit() or (s.startswith('-') and s[1:].isdigit()):
```

```
        return int(s)
    # 3. Float detection
    try:
        return float(s)
    except ValueError:
        pass
    # 4. Split on spaces *only* if the value is not enclosed in quotes
    if '␣' in s and not (s.startswith('"') and s.endswith('"')):
        return [parse_value(part) for part in s.split()]
    # 5. Strip surrounding quotes and keep inner spaces
    if s.startswith('"') and s.endswith('"'):
        return s[1:-1]
    # Fallback: return raw string
    return s

# Replace this class with your own logic implementation
class Solution:
    def solved_fun(self, *args):
        pass # TODO: Replace with your actual implementation

    def solve(self, *args) -> Union[int, str]:
        return self.solved_fun(*args)

# Main function: handles most common competitive input/output formats
def main():
    # Read all non-empty lines from stdin
    lines = [parse_value(line.strip()) for line in sys.stdin if line.strip()]

    # stdin template
    {stdin_template}

# Python script entry point
if __name__ == "__main__":
    main()
```

This Python template is designed to handle a wide variety of standard input/output formats common in programming contests. You should edit this template in three specific places when solving a new problem.

1. **import Section**

   If your solution requires additional Python libraries (e.g., math, collections, etc.), please add them to the import section at the top of the file.

   ```
   import sys, json
   from typing import List, Union
   # Add additional imports here if needed
   # e.g., from collections import defaultdict
   ```

2. **Write Your Solution Inside the Solution Class**

   Implement your logic inside the solved_fun() method. Rename the method according to the problem's context (e.g., def count_pairs(self, ...)). Make sure to update the call inside the solve() method accordingly.

   ```
   class Solution:
       def your_function_name(self, ...): # <-- Rename this function
           # Implement your solution here
           return ...

       def solve(self, *args) -> Union[int, str]:
           return self.your_function_name(*args) # <-- Make sure this matches
   ```

3. **Customize the `main()` Function's Input Parser**

You **must retain** the following line for reading input:

```
lines = [parse_value(line.strip()) for line in sys.stdin if line.strip()]
```

**Inspect the problem's input format** carefully:

- If the problem provides multiple lines of input, adjust how the input is grouped.
- If you need to handle multiple test cases, add a loop around the function calls.
- If input is structured (e.g., a matrix or multiple arrays), modify the logic to parse accordingly.

```
def main():
    lines = [parse_value(line.strip()) for line in sys.stdin if line.
        strip()]
    # stdin template # <-- Make sure the following template matches your
        problem's input format
    ...
    # print result # <-- Make sure the following template matches your
        problem's output format
```

## D.4 THE PROMPT OF THE CODING AGENT IN THE DEEP PLANNING SYSTEM

**The prompt of the Coding Agent in the Deep Planning System. (In Step 3)**

A step-by-step plan has already been created to solve this problem. Please follow the instructions below carefully:

PROBLEM

{problem}

PLANNING REPORT

{plan}

INSTRUCTIONS

1. Implement the code strictly according to the provided plan.
2. Each part of the code must clearly indicate which step in the plan it corresponds to.
3. Add meaningful in-line comments in the code to explain the logic.
4. Please provide the final complete code; do not just return the `class solution`.

{prompt_of_code_template}

- - - - - - - - - - - - - - - - - - - - - - - - - - - - - - - - - - - - - - -

OUTPUT FORMAT

CODE

```
# Your code here, following the plan step by step and using the format provided
```

- - - - - - - - - - - - - - - - - - - - - - - - - - - - - - - - - - - - - - -

**Important Instructions:**

- Strictly follow the instructions.
- Please modify the code format according to the format of **Code Format Sample**.

## D.5 DEBUGGING AGENT PROMPT IN THE DEEP PLANNING SYSTEM

---

**Debugging Agent Prompt for STD I/O Errors in the Deep Planning System. (In Step 4)**

You are given the following problem description and Python code.

- - - - - - - - - - - - - - - - - - - - - - - - - - - - - - - - - - - - - - - - - - - - -

**Problem:** {Problem}

- - - - - - - - - - - - - - - - - - - - - - - - - - - - - - - - - - - - - - - - - - - - -

**Code:**

```
{Code}
```

- - - - - - - - - - - - - - - - - - - - - - - - - - - - - - - - - - - - - - - - - - - - -

Please focus only on fixing the `main()` function.
Your task is to **rewrite only the `main()` function** so that the program correctly handles input and output according to the problem description and the design of the `Solution` class.
**Requirements:**

- Do **not** modify any part of the `Solution` class.
- You **must retain** the following line for reading input:

```
lines = [parse_value(line.strip()) for line in sys.stdin if line.
         strip()]
```

- Based on the `lines` list and the problem's input format, write the rest of the `main()` logic so that it:
  1. Correctly parses the input.
  2. Calls the appropriate method(s) from `Solution`.
  3. Prints the result in the correct format.

- - - - - - - - - - - - - - - - - - - - - - - - - - - - - - - - - - - - - - - - - - - - -

Below is a sample input/output format you should follow:

```
{sample_io}
```

Return only the modified `main()` function code. Do not return any explanations or other parts of the script.

---

**Debugging Agent Prompt for Runtime Errors in the Deep Planning System. (In Step 4)**

You are a programmer who has received a solution of a problem written in **{language}**, but it fails to compile or throws an immediate runtime error. Your task is to fix the code so that it compiles and runs correctly.

BUGGY CODE

```
{code}
```

{test_log}
**Expected Output:**
Your response must be structured as follows:

DIAGNOSIS

- Analyze the error message and buggy code.
- Identify the cause of the failure (e.g., syntax error, missing variable, etc.).
- Explain how to fix it concisely.

MODIFIED CODE

- Please keep all program comments.

- If there are errors, please also update the comments.

```
    # Corrected code with concise comments on the fix.
```

**Important Instructions:**
- Strictly follow the instructions.
- Do not change the algorithm unless necessary to fix the compile error.
- Focus on fixing syntax or execution-level issues only.
- Do not add testing code, for example, `assert` statements in your code.

---

**Debugging Agent Prompt for Wrong Answer Cases in the Deep Planning System. (In Step 4)**

You are a programmer who needs to fix a solution to a problem written in **{language}**. The code runs, but produces incorrect results on certain test cases.

PROBLEM

{problem_with_planning}

BUGGY CODE

```
    {code}
```

TEST CASE FAILURE LOG

{test_log}
**Expected Output:**

ANALYZE THE PLAN AND TEST FAILURE
- Read the problem plan and failed test case log.
- Simulate the plan's logic using one failed test input.
- Determine what the correct output should be.
- Identify what kind of mistake likely occurred (e.g., wrong algorithm, rounding error, missing constraint).

ROOT CAUSE & STRATEGY
- Clearly describe the most likely root cause.
- Do not patch old logic—redesign a **better and more reliable solution** if needed.
- Mention if precision issues suggest using better tools (e.g., `decimal` instead of `float`).

IMPROVED PLAN
- Describe an improved plan or algorithm to solve the problem more accurately and reliably.

CODE
- Please keep all program comments.
- If there are errors, please also update the comments.

```
    # Rewritten code based on improved plan, using robust logic and tools.
```

---

**Important Instructions:**
- Strictly follow the instructions.

- Do not copy old logic.
- Be open to changing tools (e.g., decimal) or redesigning the algorithm if needed.
- Do not add testing code, for example, `assert` statements in your code.

# E    ROLE OF LLMS IN THIS RESEARCH

In the preparation of this manuscript, we utilized several Large Language Models (LLMs), including OpenAI's GPT-5, Google's Gemini 2.5 Pro, and Anthropic's Claude 4 Sonnet, for two primary purposes. First, we employed these models to assist with proofreading and enhancing the grammatical clarity and readability of the text. Second, they were used as coding assistants to generate boilerplate code and utility functions within our implementation. The core research concepts, experimental design, and scientific analysis presented in this paper were conceived and executed entirely by the human authors. Our open-source code, partially developed with LLM assistance, is available to ensure full transparency and reproducibility.

