# OpenReview forum: "MACS CODER: A Multi-Agent Coding Framework for Small LMs — From Fast Thinking to Deep Planning"
_ICLR.cc/2026/Conference — ICLR 2026 Conference Withdrawn Submission_

### Official Review · Reviewer_K8Dh · 2025-10-30

**Soundness:** 3
**Presentation:** 3
**Contribution:** 2
**Rating:** 4
**Confidence:** 4

**Summary:**

The paper introduces MACS-Coder, a multi-agent code generation framework. MACS-Coder is a dual-process framework that includes a Fast-Thinking module for simple problems and a Deep-Planning module with specialized agents for complex problems. The deep planning system includes specialized agents for Planning, Coding, Debugging, and STD-IO Tools to tackle the complex problems. The framework can reach strong performance on HumanEval, MBPP, and LiveCodeBench, outperforming previous works in both accuracy and efficiency. The system also makes small open-source LLMs competitive with larger proprietary models.

**Strengths:**

* The notice of the “computation waste” phenomenon of using complete, resource-intensive stack for simple problems is practical. The idea of using the dual-process framework to dynamically allocating compute based on problem difficulty is intuitive and well-motivated.
* The system includes multiple modules with different functions. The functionality of each module is well-explained.
* The evaluation considers both the accuracy and efficiency, clearly shows the advantages of the dual-process system. The system is compared with large models and shows improvements across multiple open-source models and datasets. The ablation provides thorough ablation analyses of each component.

**Weaknesses:**

* The system uses the test cases for each problem as a criterion for multiple modules, including judging whether fast thinking module is enough, and code execution & debugging. This design makes the performance of the framework heavily depend on whether the task-specific test cases exist and the test cases’ quality. This limits the framework’s adaptability to real world questions thus weaken the contribution.
* In addition to the test cases, there are also some setups in the framework, such as adaptive prompting and STD-IO Tool activation that are manually configured per benchmark. This weakens the claims of autonomous adaptation.

**Questions:**

* Are the test cases provided in the benchmark? Is it reasonable to assume the access to the test cases and take the test cases as a necessary part and important criterion for each new problem instance? How to make sure the access and quality to test cases when tacking real world problems? If not using or not fully depending on the test cases, what are some other ways?
* “Our implementation adapts its prompting strategy based on the problem domain” in line 191. “Note that this tool is not activated for simpler benchmarks like HumanEval and MBPP, which lack complex I/O handling.” In line 208. These shows the manual configuration per benchmark. Could you discuss the performance impact if the prompt is always same and STD IO tool were always active? How would the runtime be affected then?
* How sensitive is the framework to prompt design?

---

### Official Review · Reviewer_syqN · 2025-10-31

**Soundness:** 2
**Presentation:** 3
**Contribution:** 2
**Rating:** 2
**Confidence:** 3

**Summary:**

This paper introduces a coding framework to augment the performance of open-sourced models to be close to the close-sourced larger models. I mainly concern about the novelty of the work, and the solidness of the tested benchmarks.

**Strengths:**

Refined agent workflow is great. The scores are great since the smaller open models can be close to larger reasoning models.

**Weaknesses:**

Figure 1 and Figure 2 has some contents overlapped. Figure 2 is too large and occupy too much space. The framework is not easy to follow in the schematic illustration.

Many Code benchmarks like HumanEval and MBPP own errors and the current LLMs are already near saturated in these benchmarks. Merely testing on these benchmarks are not convincing enough. Maybe consider repo-based coding like SWE-Bench.

Lack of novelty: agent framework refinement for better performance in specific tasks is not novel. Many similar works have been proposed.

**Questions:**

Have authors compared the costs? Maybe can compare with open-sourced reasoning models like DeepSeek-R1 to compare the token costs. Also compare the time latency. The agent framework requires multiple inference times, which may be unfair for comparison with one-time inference baselines.

---

### Official Review · Reviewer_WVxs · 2025-10-31

**Soundness:** 2
**Presentation:** 3
**Contribution:** 2
**Rating:** 2
**Confidence:** 3

**Summary:**

This paper introduces MACS-Coder, an efficient dual-process framework for code generation. It addresses the high resource cost of existing multi-agent systems and the performance gap between open-source models and closed-source systems. The main contribution is demonstrating that this adaptive approach significantly boosts performance and efficiency, enabling models like gpt-oss-20B to achieve SOTA results comparable to large proprietary models, thus making high-performance AI coding more accessible.

**Strengths:**

The core novelty lies in the adaptive dual-process architecture, specifically designed for efficient code generation. Dynamically switching between low-cost generation and a more complex system based on initial test success can make AI more resource-efficient.

The paper has a comprehensive evaluation, testing MACS-Coder across multiple benchmarks with various open-source models against strong baselines. Effectiveness was further demonstrated through comparisons with leading open-source and proprietary models.

The paper addresses the trade-off between performance and efficiency in AI code generation, offering a practical approach to reduce computational costs and broaden access to state-of-the-art coding capabilities.

**Weaknesses:**

In MACS-Coder, the decision to escalate to the System2 depends entirely on unit tests. While viable for benchmarks like LiveCodeBench (which provide tests), it misaligns with most real-world scenarios where tests are unavailable or co-developed with the code. This limits the framework's practical applicability, as it requires creating comprehensive unit tests in advance for any new practical task.

The LLM is only used to select an appropriate pre-built code template, not to generate it. The code templates, shown in the Appendix D, are specialized for tasks in LiveCodeBench, and cannot cover the breadth of real-world scenarios. This specialization limits the generality of the framework, as the template library would need to be manually constructed and updated for each new problem domain.

The entire workflow is a feed-forward pipeline, where issues from an early stage (e.g., low-quality unit tests, flawed planning reports) may result in errors in the final code. A reflection mechanism between stages would seem useful to mitigate this. The paper's outer planning cycles do not solve this issue. As there is no feedback or reflection between cycles, this loop is not a refinement process but a form of resampling. It re-runs the entire pipeline from scratch, rather than using the failure of the first cycle to improve the second.

The "multi-agent" claim appears to be an overstatement. The system does not actually involve different models collaborating or negotiating. Instead, it uses a single LLM backbone sequentially with different prompts, which is more accurately described as a multi-stage, multi-role pipeline. While effective, this approach does not demonstrate the more complex behaviors implied by true multi-agent systems.

**Questions:**

The automatic routing in the Debugging Agent follows a hand-coded heuristic (Section 3.2.4). This design seems questionable. Why should the first failure always be an I/O error, rather than some other common runtime error?

As shown in Table 6, when Fast Thinking is disabled, the Pass@1 (Total ACC) drops to 52.4% which is lower than 54.4% achieved by CodeSIM in Table 2. Does it imply the Fast Thinking module contributes even more than the Deep Planning module to the final performance, and the Deep Planning module is actually less effective than existing SOTA methods? Given that "Fast Thinking" is a relatively independent module, did the authors test adding this module to the strong CodeSIM baseline?

The authors state in Table 1 that they intentionally "suppress unwanted internal reasoning" by using Qwen3 in "Non-Thinking Mode" and gpt-oss with "Reasoning:Low". However, these models' stronger reasoning modes are known to perform much better on complex tasks (e.g. code generation)  [1] [2]. Could the authors provide examples of the "bad results" caused by this "unwanted reasoning"? Does this design choice imply a potential limitation, suggesting that MACS-Coder's structure may be incompatible with the internal reasoning capabilities of more powerful LLMs?

[1] Yang, An, et al. "Qwen3 technical report." arXiv preprint arXiv:2505.09388 (2025).
[2] Agarwal, Sandhini, et al. "gpt-oss-120b & gpt-oss-20b model card." arXiv preprint arXiv:2508.10925 (2025).

---

### Official Review · Reviewer_fQ5P · 2025-10-31

**Soundness:** 3
**Presentation:** 3
**Contribution:** 3
**Rating:** 6
**Confidence:** 3

**Summary:**

This paper proposes a new multi-agent framework, named MACS-Coder, for code generation with LLM. The main idea of it is quickly checking if the given problem can be easily solved (Fast Thinking). If not, then it moves to the deep planning system, consisting of planning, coding, and debugging agents, which are run in sequence. In the experiment, it is shown that the proposed approach outperforms other open LLMs with better computational efficiency, and achieves comparable performance to proprietary LLMs.

**Strengths:**

- The paper proposes a simple yet effective and novel idea of quickly checking if the given problem can be easily solved (Fast Thinking) to efficiently solve it, which is also well-motivated.
- In the experimental results, the proposed approach outperforms other open LLMs with better computational efficiency, and achieves comparable performance to proprietary LLMs.
- Writing is good and easy to read in general.

**Weaknesses:**

- Except the fast thinking, the novelty of other components seem incremental compared to other multi-agent code generation approaches like MapCoder.
- My main concern is that it was not well demonstrate which component contributes to the performance gain and computational efficiency. Even though the authors refer to the appendix for ablation study, this is quite important and should be in the main body of the paper.
- The paper lacks in-depth analysis. e.g., when the fast thinking is effective and when it is not? What are the failure cases? Also qualitative analysis like generated sample comparison is missing.
- Even though authors present extensive evaluation, it lacks one more desirable experiment -- using powerful proprietary LLMs as its backbone to demonstrate the upper bound of its performance.

**Questions:**

Please see and address the weaknesses above.

---

### Note · Authors · 2025-12-03

I have read and agree with the venue's withdrawal policy on behalf of myself and my co-authors.